# Surrounding greenness is associated with lower risk and burden of low birth weight in Iran

Siqi Luo [1,4], Yaqi Wang[1,4], Fatemeh Mayvaneh [2] ✉, Helder Relvas[3], Mohammad Baaghideh[2], Kai Wang[1], Yang Yuan[1], Zhouxin Yin[1] & Yunquan Zhang [1] ✉

The nexus between prenatal greenspace exposure and low birth weight (LBW) remains largely unstudied in low- and middle-income countries (LMICs). We investigated a nationwide retrospective cohort of 4,021,741 live births (263,728 LBW births) across 31 provinces in Iran during 2013–2018. Greenness exposure during pregnancy was assessed using satellite-based normalized difference vegetation index (NDVI) and enhanced vegetation index (EVI). We estimated greenness-LBW associations using multiple logistic models, and quantified avoidable LBW cases under scenarios of improved greenspace through counterfactual analyses. Association analyses provide consistent evidence for approximately L-shaped exposure-response functions, linking 7.0–11.5% declines in the odds of LBW to each 0.1-unit rise in NDVI/EVI with multiple buffers. Assuming causality, 3931–5099 LBW births can be avoided by achieving greenness targets of mean NDVI/EVI, amounting to 4.4–5.6% of total LBW births in 2015. Our findings suggest potential health benefits of improved greenspace in lowering LBW risk and burden in LMICs.

Accumulating studies have linked greenness with an array of health improvements, including decreased mortality, enhanced cognitive functions, and positive birth outcomes[1]. Pregnancy is a susceptible time period for environmental exposures, and adverse birth outcomes are known to be associated with neonatal morbidity and mortality, as well as life-long consequences such as stunted growth and mental retardation[2]. Given the well-defined exposure period during pregnancy, birth outcomes may be particularly helpful in identifying potential health effects of greenness[3].

To date, a number of population-based studies have examined the associations between greenness and birth outcomes[1], while the overall evidence on low birth weight (LBW) remains less consistent[4]. Notably, there still existed considerable research gaps in prior greenness-LBW investigations. First, one in seven live births globally suffered from LBW in 2015[5], of which ~90% occurred in low- and middle-income countries (LMICs). Despite this fact, the majority of existing greenness-LBW studies have been conducted in high-income countries (HICs), leaving an extensive scarcity of evidence in LMICs[6]. Second, only sporadic studies have provided national evidence using millions of live birth records spanning the entire country[7,8], thus making it inadequate and not convictive for valuing greening efforts as health-promoting measures in public policy making. Third, no prior analysis has quantified the potential benefit of reducing LBW in LMICs by achieving the well-defined target of green space coverage. To achieve the Global Nutrition Targets 2025 for LBW—a 30% reduction in the number of LBW babies by 2025[9], it is greatly essential to clarify to what extent the enhancement of vegetation density can contribute to alleviating the LBW burden in LMICs.

[1]Hubei Province Key Laboratory of Occupational Hazard Identification and Control, School of Public Health, Institute of Social Development and Health Management, Wuhan University of Science and Technology, 430065 Wuhan, China. [2]Faculty of Geography and Environmental Sciences, Hakim Sabzevari University, Sabzevar 9617916487 Khorasan Razavi, Iran. [3]CESAM & Department of Environment and Planning, University of Aveiro, 3810–193 Aveiro, Portugal. [4]These authors contributed equally: Siqi Luo, Yaqi Wang. ✉e-mail: fmayvaneh@yahoo.com; YunquanZhang@wust.edu.cn

In this study, we conceived a nationwide observational study of ~4 million mother–infant pairs, utilizing birth records from 2013 to 2018 spanning 31 provinces in Iran. Our primary purpose was to derive the nationally representative exposure–response (E–R) relationship between greenness exposure during pregnancy and the risk of LBW in Iran; a secondary aim was to quantify the avoidable burden of LBW attributable to improved greenness under counterfactual exposure scenarios.

## Results

### Summary characteristics

We investigated birth outcomes of LBW (defined as birth weight below 2500 grams, regardless of gestational age [GA]) and term low birth weight (TLBW, birth weight below 2500 grams for pregnancies with at least 37 completed weeks of gestation). By excluding 47,102 ineligible birth records (Fig. 1), we analyzed 4,021,741 Iranian mother–infant pairs from 2013 to 2018. Among them, 263,728 (6.6%) were LBW and 121,852 (3.0%) were TLBW (Table 1). The average (±standard deviation, SD) birth weights of LBW and TLBW infants were 2024.8 ± 479.2 and 2196.7 ± 407.9 grams, respectively. About a quarter of mothers with LBW/TLBW infants were village residents, and nearly half were ≤25 or >35 years old and below high school education.

Greenness exposure during pregnancy was measured using the satellite-based normalized difference vegetation index (NDVI) and enhanced vegetation index (EVI) with multiple buffers (500-m, 1000-m, 2000-m and 3000-m). Compared with mothers with normal birthweight infants, mothers with LBW/TLBW infants experienced consistently lower greenness during pregnancy within 500-m to 3000-m buffers. For instance, mothers with LBW infants were exposed to mean NDVI 0.287–0.292 and EVI 0.207–0.210, while mothers with normal birthweight infants were exposed to mean NDVI 0.315–0.319 and EVI 0.227–0.230, respectively.

### Greenness–LBW associations

We investigated the associations of greenness exposures with LBW and TLBW in multiple logistic regression models with sequential adjustments (Fig. 2). Largely consistent decreases in risks of LBW and TLBW were identified across models and greenness indices, while the associations were slightly attenuated when controlling for environmental factors or narrowing the buffers of exposures. Specifically, in a fully adjusted model, per 0.1-unit increase in $NDVI_{-3000m}$ and $EVI_{-3000m}$ were associated with odds ratios (ORs) of 0.911 (95% confidence interval [CI]: 0.908–0.914) and 0.885 (0.881–0.889) for LBW, 0.899 (0.895–0.902) and 0.871 (0.866–0.875) for TLBW, respectively. The associations between

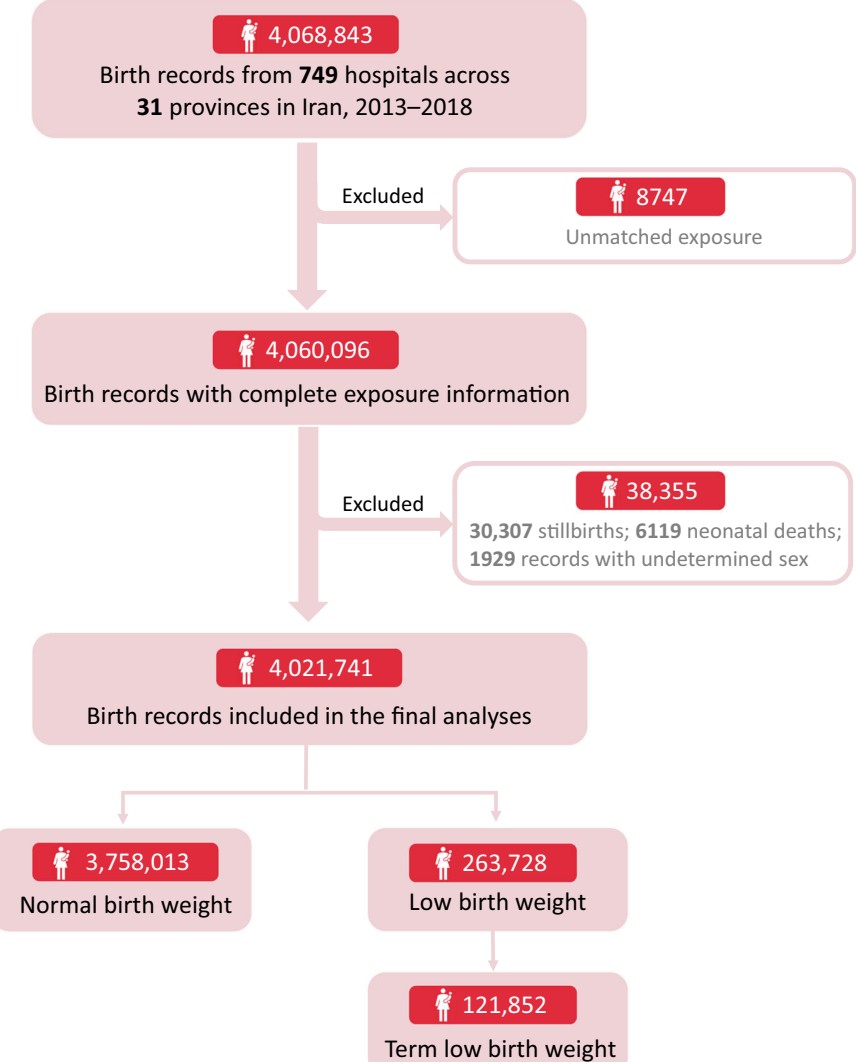

**Fig. 1 | Flow chart of the selection criteria for analyses.** For the purpose of investigating the association between greenspace and low birth weight, we excluded 47,102 ineligible birth records (e.g., stillbirths and neonatal deaths), a total of 4,021,741 mother–infant pairs were included in the final analysis.

**Table 1 | Descriptive characteristics of participants**

| Characteristic | No. (%) | | | |
|---|---|---|---|---|
| | Total | Normal birth weight | Low birth weight | Term low birth weight |
| Subjects | 4,021,741 | 3,758,013 (93.4) | 263,728 (6.6) | 121,852 (3.0) |
| Birth weight (grams, mean ± SD) | 3176.2 ± 503.2 | 3257.0 ± 394.0 | 2024.8 ± 479.2 | 2196.7 ± 407.9 |
| Gestational age (weeks, mean ± SD) | 38.5 ± 1.7 | 38.7 ± 1.2 | 35.4 ± 3.5 | 38.3 ± 1.0 |
| *Infant sex* | | | | |
| Male | 2,074,664 (51.6) | 1,951,484 (51.9) | 123,180 (46.7) | 51,039 (41.9) |
| Female | 1,947,077 (48.4) | 1,806,529 (48.1) | 140,548 (53.3) | 70,813 (58.1) |
| *Season of birth* | | | | |
| Spring | 976,828 (24.3) | 909,836 (24.2) | 66,992 (25.4) | 30,936 (25.4) |
| Summer | 1,140,987 (28.4) | 1,066,848 (28.4) | 74,139 (28.1) | 34,309 (28.2) |
| Autumn | 980,745 (24.4) | 917,468 (24.4) | 63,277 (24.0) | 28,891 (23.7) |
| Winter | 923,181 (23.0) | 863,861 (23.0) | 59,320 (22.5) | 27,716 (22.7) |
| *Number of fetuses* | | | | |
| Singleton | 3,976,387 (98.9) | 3,738,825 (99.5) | 237,562 (90.1) | 115,228 (94.6) |
| Multiple gestations | 45,354 (1.1) | 19,188 (0.5) | 26,166 (9.9) | 6624 (5.4) |
| *Maternal age, yrs* | | | | |
| ≤25 | 1,453,336 (36.1) | 1,358,339 (36.1) | 94,997 (36.0) | 46,289 (38.0) |
| (25, 35] | 2,133,831 (53.1) | 1,999,070 (53.2) | 134,761 (51.1) | 61,375 (50.4) |
| >35 | 434,574 (10.8) | 400,604 (10.7) | 33,970 (12.9) | 14,188 (11.6) |
| *Parity* | | | | |
| 0 | 1,670,615 (41.5) | 1,543,838 (41.1) | 126,777 (48.1) | 59,097 (48.5) |
| ≥1 | 2,351,126 (58.5) | 2,214,175 (58.9) | 136,951 (51.9) | 62,755 (51.5) |
| *Residence* | | | | |
| City | 3,097,265 (77.0) | 2,897,059 (77.1) | 200,206 (75.9) | 90,849 (74.6) |
| Village | 924,476 (23.0) | 860,954 (22.9) | 63,522 (24.1) | 31,003 (25.4) |
| *Education attainment* | | | | |
| Below high school | 1,927,541 (47.9) | 1,791,513 (47.7) | 136,028 (51.6) | 63,459 (52.1) |
| High school | 1,206,248 (30.0) | 1,133,219 (30.2) | 73,029 (27.7) | 34,050 (27.9) |
| College or above | 887,952 (22.1) | 833,281 (22.2) | 54,671 (20.7) | 24,343 (20.0) |
| *Nationality* | | | | |
| Iranian | 3,873,207 (96.3) | 3,619,299 (96.3) | 253,908 (96.3) | 117,106 (96.1) |
| Non-Iranian | 148,534 (3.7) | 138,714 (3.7) | 9820 (3.7) | 4746 (3.9) |
| *Delivery type* | | | | |
| Cesarean section | 2,059,655 (51.2) | 1,903,604 (50.7) | 156,051 (59.2) | 66,019 (54.2) |
| Vaginal delivery | 1,962,086 (48.8) | 1,854,409 (49.3) | 107,677 (40.8) | 55,833 (45.8) |
| *Delivery complications* | | | | |
| Absent | 3,914,965 (97.3) | 3,657,444 (97.3) | 257,521 (97.6) | 119,018 (97.7) |
| Present | 106,776 (2.7) | 100,569 (2.7) | 6207 (2.4) | 2834 (2.3) |
| *Diabetes* | | | | |
| Absent | 3,919,203 (97.5) | 3,663,529 (97.5) | 255,674 (96.9) | 119,238 (97.9) |
| Present | 102,538 (2.5) | 94,484 (2.5) | 8054 (3.1) | 2614 (2.1) |
| *Chronic hypertension* | | | | |
| Absent | 3,959,776 (98.5) | 3,704,473 (98.6) | 255,303 (96.8) | 119,196 (97.8) |
| Present | 61,965 (1.5) | 53,540 (1.4) | 8425 (3.2) | 2656 (2.2) |
| *Environmental exposures* | | | | |
| $NDVI_{500m}$ | 0.313 ± 0.21 | 0.315 ± 0.21 | 0.287 ± 0.21 | 0.276 ± 0.20 |
| $NDVI_{1000m}$ | 0.314 ± 0.20 | 0.316 ± 0.20 | 0.288 ± 0.20 | 0.276 ± 0.20 |
| $NDVI_{2000m}$ | 0.315 ± 0.20 | 0.317 ± 0.20 | 0.290 ± 0.19 | 0.278 ± 0.19 |
| $NDVI_{3000m}$ | 0.317 ± 0.19 | 0.319 ± 0.19 | 0.292 ± 0.19 | 0.280 ± 0.19 |
| $EVI_{500m}$ | 0.225 ± 0.16 | 0.227 ± 0.16 | 0.207 ± 0.15 | 0.200 ± 0.15 |
| $EVI_{1000m}$ | 0.226 ± 0.15 | 0.227 ± 0.15 | 0.207 ± 0.15 | 0.199 ± 0.15 |
| $EVI_{2000m}$ | 0.227 ± 0.15 | 0.228 ± 0.15 | 0.209 ± 0.14 | 0.201 ± 0.14 |
| $EVI_{3000m}$ | 0.229 ± 0.14 | 0.230 ± 0.14 | 0.210 ± 0.14 | 0.202 ± 0.14 |
| $PM_{2.5}$, μg/m³ | 41.1 ± 12.6 | 41.1 ± 12.6 | 41.9 ± 12.9 | 42.2 ± 12.4 |
| Temperature, °C | 16.7 ± 5.8 | 16.7 ± 5.7 | 17.0 ± 6.3 | 17.3 ± 6.0 |
| Relative humidity, % | 43.5 ± 13.3 | 43.5 ± 13.2 | 42.4 ± 13.7 | 41.6 ± 13.1 |

Delivery complications refer to a range of medical issues and problems that can occur during the process of delivery, including one or more injuries of the parturient canal, placenta abruption, premature rupture of membranes, shoulder dystocia, uterine rupture at delivery, etc. Diabetes is defined as type 1 or type 2 diabetes diagnosed prior to conception (without gestational diabetes). Chronic hypertension is defined as pre-pregnancy hypertension (≥140/90 mmHg) or hypertension before 20 weeks of pregnancy. The sum of percentages from multiple subgroups may not equal 100% exactly due to rounding-off numbers.
*SD* standard deviation, *NDVI* normalized difference vegetation index, *EVI* enhanced vegetation index, $PM_{2.5}$ fine particulate matter.

**a**

| | Model 1 | Model 2 | Model 3 | Model 4 |
|---|---|---|---|---|
| NDVI_500m | 0.933 (0.931–0.935) | 0.923 (0.921–0.925) | 0.917 (0.914–0.919) | 0.930 (0.927–0.933) |
| NDVI_1000m | 0.930 (0.928–0.932) | 0.919 (0.916–0.921) | 0.911 (0.909–0.914) | 0.923 (0.920–0.926) |
| NDVI_2000m | 0.927 (0.925–0.929) | 0.914 (0.912–0.916) | 0.906 (0.904–0.909) | 0.916 (0.914–0.919) |
| NDVI_3000m | 0.924 (0.922–0.926) | 0.910 (0.908–0.913) | 0.902 (0.900–0.905) | 0.911 (0.908–0.914) |
| EVI_500m | 0.916 (0.913–0.918) | 0.902 (0.899–0.905) | 0.894 (0.891–0.897) | 0.913 (0.910–0.917) |
| EVI_1000m | 0.909 (0.907–0.912) | 0.894 (0.891–0.897) | 0.886 (0.883–0.889) | 0.904 (0.900–0.908) |
| EVI_2000m | 0.904 (0.901–0.906) | 0.887 (0.884–0.890) | 0.877 (0.874–0.880) | 0.893 (0.890–0.897) |
| EVI_3000m | 0.899 (0.897–0.902) | 0.881 (0.878–0.884) | 0.871 (0.868–0.874) | 0.885 (0.881–0.889) |

**b**

| | Model 1 | Model 2 | Model 3 | Model 4 |
|---|---|---|---|---|
| NDVI_500m | 0.909 (0.906–0.911) | 0.910 (0.907–0.912) | 0.902 (0.899–0.905) | 0.921 (0.918–0.924) |
| NDVI_1000m | 0.902 (0.899–0.905) | 0.903 (0.901–0.906) | 0.895 (0.892–0.898) | 0.913 (0.910–0.917) |
| NDVI_2000m | 0.897 (0.894–0.900) | 0.898 (0.895–0.901) | 0.889 (0.886–0.891) | 0.906 (0.902–0.909) |
| NDVI_3000m | 0.892 (0.889–0.895) | 0.893 (0.890–0.896) | 0.883 (0.880–0.886) | 0.899 (0.895–0.902) |
| EVI_500m | 0.885 (0.881–0.888) | 0.886 (0.882–0.890) | 0.877 (0.873–0.880) | 0.902 (0.898–0.907) |
| EVI_1000m | 0.875 (0.871–0.879) | 0.876 (0.873–0.880) | 0.866 (0.862–0.870) | 0.892 (0.887–0.896) |
| EVI_2000m | 0.866 (0.863–0.870) | 0.868 (0.864–0.871) | 0.856 (0.852–0.860) | 0.880 (0.876–0.885) |
| EVI_3000m | 0.859 (0.855–0.863) | 0.860 (0.856–0.864) | 0.848 (0.844–0.852) | 0.871 (0.866–0.875) |

OR
0.85   0.87   0.89   0.91   0.93

**Fig. 2 | Model-specific odds ratios and 95% confidence intervals of LBW/TLBW for per 0.1-unit increase in vegetation indices within multiple buffers.** Multi-model associations of LBW (**a**) and TLBW (**b**) with NDVI or EVI. Model 1 was a crude model without covariate adjustment; Model 2 was adjusted for maternal age, infant sex, and gestational age; Model 3 was adjusted for covariates in maternal demographic characteristics and fetal variables; Model 4 (fully-adjusted model) was adjusted for covariates in maternal demographic characteristics, fetal variables, and environmental factors. LBW low birth weight, TLBW term low birth weight, NDVI normalized difference vegetation index, EVI enhanced vegetation index, OR odds ratio.

greenness and LBW/TLBW remained greatly robust to sensitivity analyses based on various exclusion criteria and additional adjustments for socioeconomic status (SES) variables (Table S1). Also, analyses by exposure quartiles showed that increased greenness exposures during pregnancy were associated with lower odds of LBW and TLBW (*P* for trend <0.001, Table 2). For instance, compared with the lowest NDVI_3000m quartile, individuals exposed to the highest NDVI_3000m quartile shared significantly lower ORs of 0.961 (0.960–0.963) for LBW and 0.956 (0.955–0.958) for TLBW, respectively. Comparable risk estimates were seen in analyses using different greenness indices and buffers (Tables S2 and S3).

E–R relationships between LBW/TLBW and greenness exposures within multiple buffers were outlined in Fig. 3. We observed strikingly non-linear associations (*P* < 0.001) between greenness exposures and LBW/TLBW, suggesting consistent evidence for L-shaped curves under different greenness indices and buffers (Fig. S1). Greenness improvement at relatively low NDVI/EVI exposures was substantially associated with decreased risks of LBW, whereas greatly attenuated benefits were identified across high greenness levels. For instance, within a 3000-m buffer, the risk of TLBW was sharply diminished when NDVI exposures were lower than 0.4, while progress almost stagnated for higher exposures.

Estimates of subgroup analyses were presented for the associations between LBW/TLBW and greenness within a 3000-m buffer, stratified by age, education attainment and residence (Fig. 4). Consistent with other buffers (Figs. S2–S4), significantly larger decreases in greenness-related risks were estimated among several subpopulations. The associations between NDVI_3000m and LBW were stronger among younger (≤25 years, OR: 0.895, 95% CI: 0.890–0.900), less educated (below high school, 0.897, 0.893–0.901), and village-dwelling (0.893, 0.887–0.898) mothers. Similar results were also observed for TLBW and EVI_3000m (Table S4).

**Estimation of avoidable LBW assuming causality**
Under predefined causality and counterfactual scenarios of improved NDVI or EVI, we estimated greenness-related attributable fractions (AFs) and avoidable numbers (ANs) of LBW/TLBW by linking the E–R functions to spatially resolved estimates of live births, and maternal greenness exposures (Fig. 5). Higher proportions but lower ANs of TLBWs were attributable to greenness compared to those of LBWs. The AFs and ANs varied appreciably across buffers, with the highest estimates for 3000-m buffers and the lowest estimates for 500-m buffers. In 2015, 3931–5099 LBWs and 2173–2697 TLBWs could be avoided by achieving greenness targets of mean NDVI/EVI with buffers between 500 m and 3000 m during 2013–2018, representing 4.4–5.6% of total LBWs and 5.6–6.9% of total TLBWs in Iran (Table S5). The estimates of avoidable LBWs were roughly twice as high as those of avoidable TLBWs.

**Table 2 | Associations of LBW and TLBW with quartiles of greenness exposure within a 3000-m buffer**

| Quantiles | Median | LBW | | TLBW | |
|---|---|---|---|---|---|
| | | OR (95% CI) | P for trend[a] | OR (95% CI) | P for trend[a] |
| *NDVI* | | | | | |
| Q1 | 0.111 | 1 (Ref.) | <0.001 | 1 (Ref.) | <0.001 |
| Q2 | 0.242 | 0.984 (0.982–0.985) | | 0.976 (0.975–0.978) | |
| Q3 | 0.338 | 0.970 (0.969–0.972) | | 0.963 (0.961–0.965) | |
| Q4 | 0.512 | 0.961 (0.960–0.963) | | 0.956 (0.955–0.958) | |
| *EVI* | | | | | |
| Q1 | 0.074 | 1 (Ref.) | <0.001 | 1 (Ref.) | <0.001 |
| Q2 | 0.176 | 0.984 (0.982–0.985) | | 0.976 (0.974–0.977) | |
| Q3 | 0.242 | 0.972 (0.971–0.974) | | 0.966 (0.964–0.968) | |
| Q4 | 0.382 | 0.963 (0.961–0.964) | | 0.958 (0.956–0.960) | |

These associations are derived from the fully adjusted model, where covariates include maternal demographic characteristics, fetal variables, and environmental factors. The statistical tests are two-sided.
*OR* odds ratio, *CI* confidence interval, *NDVI* normalized difference vegetation index, *EVI* enhanced vegetation index, *LBW* low birth weight, *TLBW* term low birth weight.
[a]Tested by introducing the median value of each quartile as a continuous variable in the logistic regression model.

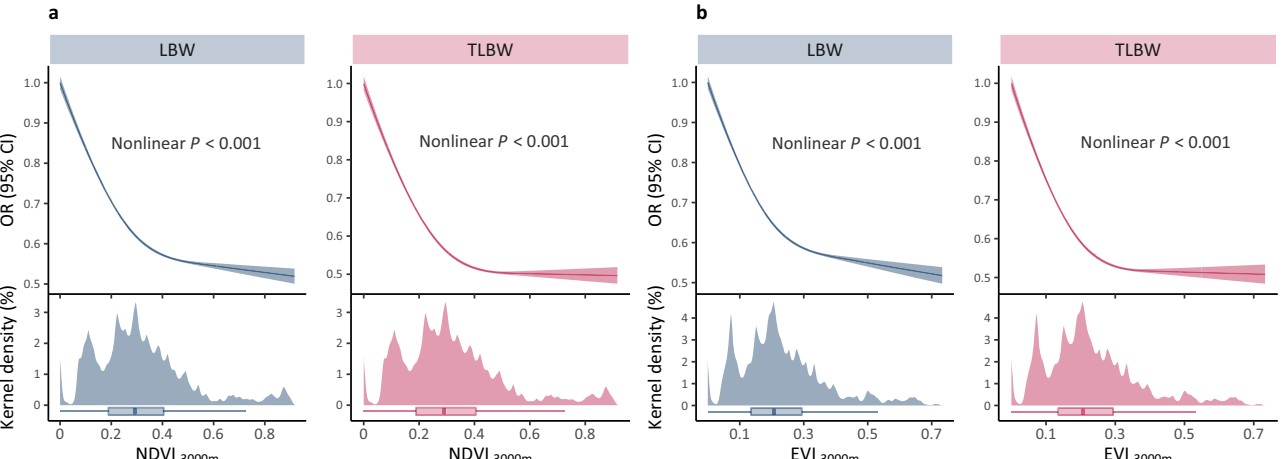

**Fig. 3 | Exposure–response curves for greenness-LBW/TLBW associations.** Relationships between LBW/TLBW and NDVI (**a**) or EVI (**b**) within a 3000-m buffer. The curves stemmed from the fully adjusted model (including maternal demographic characteristics, fetal variables, and environmental factors) are fitted using restricted cubic splines function with three knots placed at the 10th, 50th, and 90th percentiles of greenness exposure distribution. The solid lines in the upper panels represent the point estimates of LBW/TLBW risks compared to 0, while the shaded bands represent their 95% confidence intervals. In the lower panels, we have displayed the kernel density curves and box plots to illustrate the distributions of NDVI and EVI exposures. In the box plots, the center bars are medians; the box bounds indicate ranges from 25th to 75th; whiskers refer to minima and maxima, and extreme values beyond the 75th percentile plus 1.5 times IQR are not displayed. The nonlinear is examined by likelihood-ratio test, and statistical tests are two-sided. OR odds ratio, CI confidence interval, LBW low birth weight, TLBW term low birth weight, NDVI normalized difference vegetation index, EVI enhanced vegetation index.

## Discussion

This national study investigated the association between greenness exposure during pregnancy and LBW in LMICs. By involving ~4 million Iranian live births, our study provided robust evidence that elevated greenness was associated with decreased risks of LBW and TLBW. Stratified analyses revealed potential heterogeneities in greenness-LBW associations between subpopulations, suggesting significantly greater greenness-associated benefits among younger, less educated, and village-dwelling mothers. These findings from association analyses suggested the potential health benefits of improved greenness in lowering LBW risk and burden in LMICs.

The current study associated a decreased LBW risk of 7.0% (95% CI: 6.7–7.3) with a per 0.1-unit increase in NDVI$_{-500m}$. This estimate was slightly lesser than the pooled risk reduction of 10.0% (1.0–17.0) in a recent meta-analysis of 10 observational studies[4], despite substantial heterogeneity among included studies (OR range: 0.55–1.03). Such a discrepancy in effect magnitude could be largely predictable, given

that included studies in the pooled analysis were mostly conducted in HICs (9/10). In several studies focused on populations in LMICs, considerable uncertainties also remained with regard to greenness-LBW associations. Most recently, two regional studies in southern China involving 11,258 and 16,184 live births reported reduced risks of 0.69 (0.56–0.85)[10] and 0.91 (0.83–0.99)[11] per 0.1-unit increase in NDVI$_{-500m}$, respectively, whereas a null association of NDVI$_{-500m}$ with LBW was found among 18,655 live births in northern China (0.96 [0.78–1.19])[12]. Great diversity in study demographics and vegetation cover may partially interpret these mixed findings on the direction of associations and the magnitude of effect sizes. Our findings provided national evidence on greenness-LBW association in LMICs, suggesting a possibly effective intervention measure for reaching the Global Nutrition Targets 2025 for LBW in these regions[9]. More nationally representative or multi-country investigations are urgently needed in LMICs as reference bases for how much increase in vegetation density would be practically appropriate in areas with high LBW prevalence.

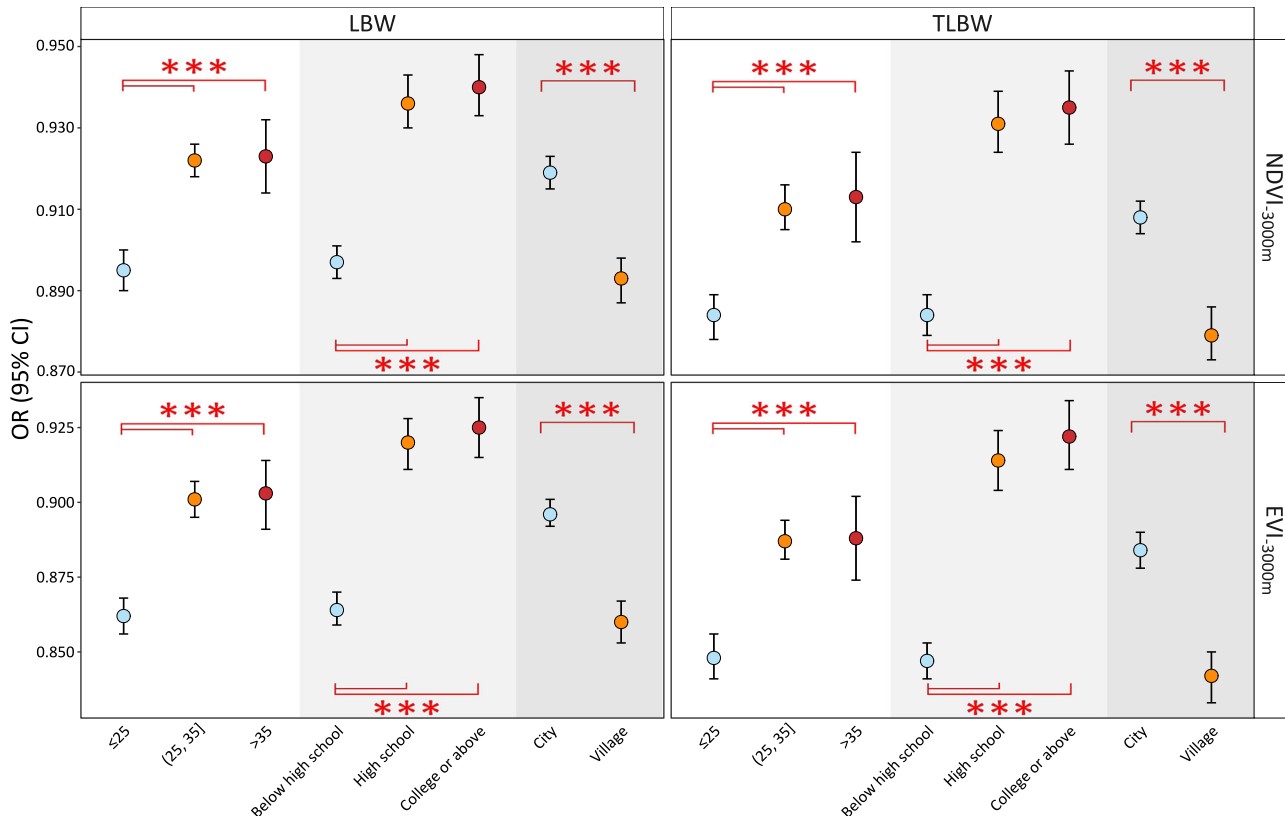

**Fig. 4 | Subgroup-specific OR (95% CI) estimates of LBW and TLBW associated with a 0.1-unit increase in NDVI and EVI within a 3000-m buffer.** These estimates are derived from the fully adjusted model, where covariates include maternal demographic characteristics, fetal variables, and environmental factors. Error bars represent 95% confidence intervals. Effect heterogeneity between subgroups is examined through a fixed effect meta-regression method, and the statistical tests are two-sided. The exact sample sizes for subgroups are exhibited in Table 1. *** indicates $P < 0.001$ for effect heterogeneity between subgroups. OR odds ratio, CI confidence interval, LBW low birth weight, TLBW term low birth weight, NDVI normalized difference vegetation index, EVI enhanced vegetation index.

In this study, we observed approximately L-shaped E–R curves between greenness and LBW, suggesting a greater reduction in LBW risk at the lower greenness range. This interesting finding was partially echoed by prior investigations on several fetal parameters, despite that heterogeneous E–R functions were still identified[10,13,14]. For instance, inverse U-shaped curves were identified for relationships between NDVI within a 500-m buffer and birth weight, birth height, and GA based on China Maoming Birth Cohort[10], indicating that a moderate greenness level may support more favorable fetal parameters. According to the analysis from the Massachusetts Birth Registry data[15], NDVI-birthweight association showed an S pattern and significantly larger increments in birth weight were found in low NDVI ranges of 0.25–0.50 than NDVI 0.50–0.75, with an estimate of 6.7 grams (95% CI: 5.2–8.2) versus 2.1 grams (95% CI: 0.1–4.1) per 0.1-unit rise in $NDVI_{-250m}$. By involving 3 million births, the Texas Vital Statistics program found participants residing in $NDVI_{-250m}$ between 0.37 and 0.45 had the most increase in birth weight (2.7 grams) compared to those residing in $NDVI_{-250m}$ above 0.45[14]. This evidence from both developed and less developed countries highlighted the potential for substantial health benefits from improvements in vegetation density at low-level greenness areas, which may help reduce socioeconomic health inequalities.

As the "equigenic environments" theory suggested, greenness might disrupt the conventional transition of socioeconomic adversity to greater health risk and may be important in filling the pervasive health gap between more and less advantaged persons[3]. Less years of education are widely believed to relate to lower SES[16]. Though several studies found benefits irrespective of SES or more benefits in participants belonging to the highest SES groups[7,13,15], the present study and more existing studies reported stronger associations between greenness and birth outcomes among mothers with low education level or low SES[17,18], who may be highly deprived of green environment[19]. Emerging evidence suggested inconsistent findings for urban–rural disparity in associations of vegetation with birth outcomes[4]. In contrast to prior investigations[10,20], our study found stronger associations among village residents rather than urban dwellers. This inconsistency may be explained by the variation in the criteria used to classify urban and rural areas between studies, and urban residents in some regions may have easier access to higher quality of green spaces as NDVI or EVI cannot tell the type of green spaces[16]. Compared to residential and educational disparities, age differences in vegetation-birth associations are less discussed in existing studies. In line with a population-based Chinese birth cohort analysis on preterm delivery, we identified significantly greater benefits of greenness in lowering the LBW risks for younger mothers (≤25 years old)[16]. Younger mothers may face unstable financial income and greater survival stress, usually implying a lower SES. To fill this health gap due to SES inequalities, increasing density of vegetation could be an effective intervention and should be prioritized in the most socioeconomically deprived communities[18,21,22].

Through counterfactual analyses, prior studies have reported the alleviated burden of death related to greenspace improvement. Small-scale studies have been pioneered in individual cities; in Philadelphia, USA, 403 (95% CI: 298–618) deaths would be prevented annually for reaching the goal of 30% tree canopy cover in each of the city's neighborhoods[23], and in Barcelona, Spain, increasing percentage of green area from 6.5% to 19.6% was estimated to avoid 60 (0–119) deaths in 2010[24]. A larger-scale study encompassing 31 European cities found that meeting the WHO recommendation of at least 0.5 hectares

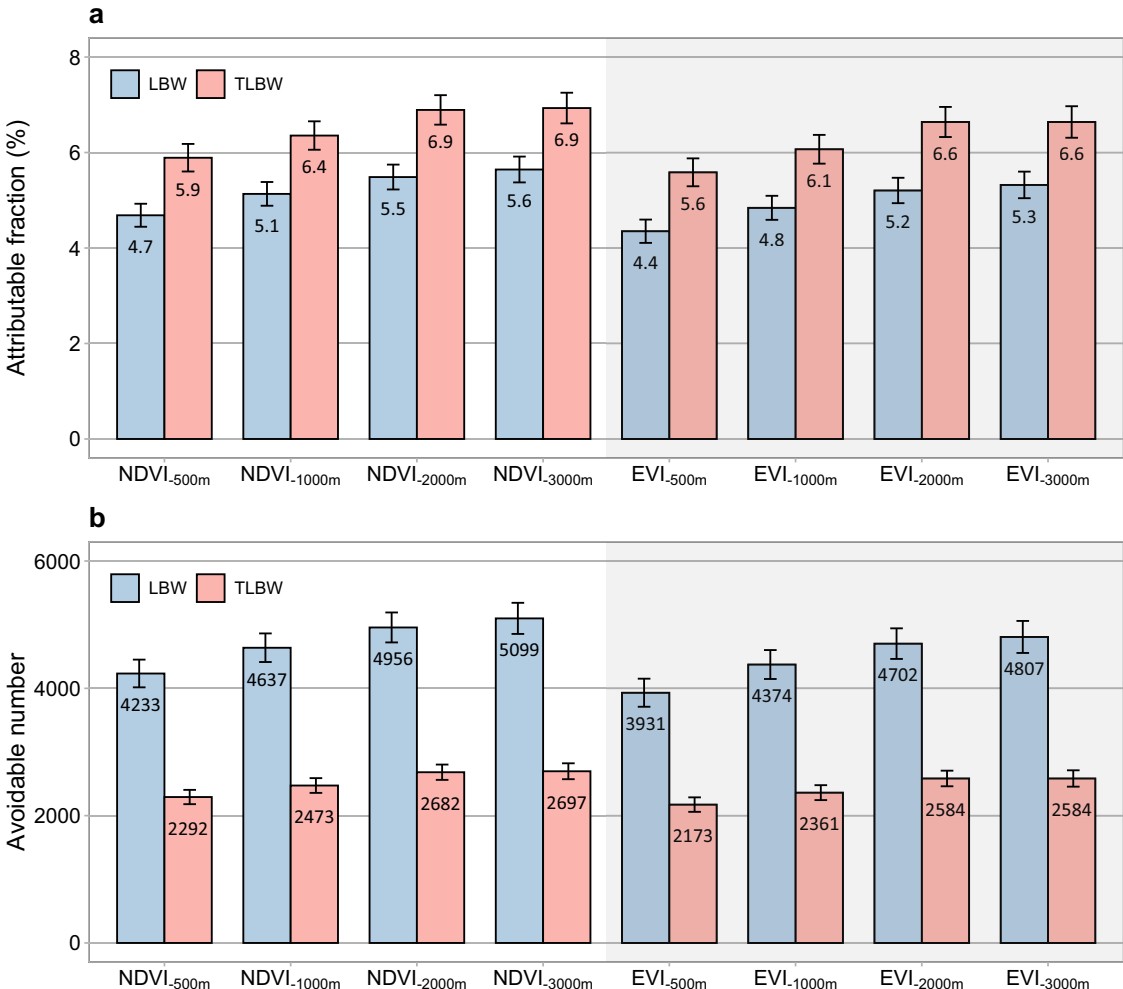

**Fig. 5 | Burden estimates of LBW and TLBW across Iran in 2015.** Attributable fraction (**a**) and avoidable number (**b**) by achieving greenness targets of mean NDVI/EVI within multiple buffers. Error bars represent 95% confidence intervals.

LBW low birth weight, TLBW term low birth weight, NDVI normalized difference vegetation index, EVI enhanced vegetation index.

of green spaces within a 300-m distance could prevent 42,968 (32,296–64,177) deaths in 2015[25]. These analyses emphasized the great importance of green space in reducing premature deaths and prolonging life expectancy in developed countries[1]. Quantitative contribution of greenness improvement to reducing adverse birth outcomes remains unstudied across the globe, whereas a number of existing studies in LMICs have investigated the potential health benefits of weakening exposure to adverse environmental factors, such as ambient fine particular matter ($PM_{2.5}$) pollution. For instance, a modeling analysis of Demographic and Health Surveys (DHS) data estimated a total of 0.83 (0.54–1.08) million stillbirths attributable to $PM_{2.5}$ exposure exceeding the counterfactual exposure of 10 $\mu g/m^3$, amounting to 39.7% (26.1–50.8) of total stillbirths across 137 countries in 2015[26]. Using 2.4 $\mu g/m^3$ as a reference, 1018 (718–1289) thousand and 280 (196–358) thousand $PM_{2.5}$-associated preterm births were estimated to be avoided annually during 1990–2019 in India and China, respectively[27]. Compared with ambient $PM_{2.5}$ exposure, huge research gaps still exist in greenness-attributable assessment, primarily owing to the lack of nationwide evidence regionally and globally. Assuming causality for greenness-LBW associations, our study estimated that, in Iran, 3931–5099 LBWs could be avoided by achieving targets of mean NDVI/EVI with various buffers, representing 4.4–5.6% of total LBWs in 2015. Considering that greenness may alleviate $PM_{2.5}$-related risks of adverse outcomes[28], co-improvements of increased vegetation and

clean air could helpfully accelerate the progress of Global Nutrition Targets for LBW in 2025.

Some limitations of our study need to be noted. First, individual maternal greenness exposures were evaluated according to locations of delivery hospitals rather than residential communities, yet robust findings from buffering analyses (500–3000 m) may reduce the influence of such exposure bias to some extent because pregnant women mostly give birth to a hospital not too far from home[29]. Second, NDVI and EVI may exhibit limitations in fully characterizing the maternal experience of being exposed to trees and plants because these indicators do not reflect the type and quality of vegetation[30]. However, NDVI and EVI can still provide guidance in less developed areas as some of the few widely validated and globally covered vegetation indicators. Third, unmeasured confounding (e.g., body-mass index and smoking status) was likely from the lack of control for potential individual-level risk factors of LBW. However, sensitive analyses via additionally adjusting for provincial-level SES variables provided comparable risk estimates with our primary results. Also, prior cohort studies from Asia[11] and Europe[31] reported similar greenness-LBW associations between unadjusted and multivariable-adjusted analyses adjusted for various covariates including lifestyle factors such as maternal smoking status. Such evidence suggested a limited influence of residual confounding on our findings. Fourth, our attributable assessments were on the basis of E–R functions derived from

association analysis instead of the causal framework, estimates of avoidable LBW should thus be interpreted with caution.

In summary, our study added nationally robust evidence for decreased risks of LBW in association with greenness exposures in different buffers among 4 million Iranian newborns. Approximately L-shaped E−R functions with slopes being steeper at low greenness levels were identified between greenness and LBW. Babies of younger, less educated, and village-dwelling mothers had a greater reduction in the risk of LBW associated with an increase in surrounding greenspace. Our quantitative estimation for avoidable LBW and TLBW in Iran, based on association analyses and counterfactual framework, suggested underlying implications for national and global assessments of the disease burden of adverse birth outcomes associated with low-greenness exposures and may provide valuable information to address inequalities in LBW risk and to promote environmental justice in LMICs. Future assessments from multiple countries, utilizing sophisticated designs and causal analysis, are greatly warranted to validate findings from this study.

## Methods
### Study population
This nationwide retrospective birth cohort included 4,068,843 birth records from 749 hospitals (Figure S5) across 31 Iranian provinces between January 2013 and December 2018. These birth data were derived from the Ministry of Health and Medical Education, which owns and operates the largest healthcare delivery network in Iran. The individual records comprised information on maternal socio-demographic characteristics (e.g., age at birth, parity, education attainment, health status), infant characteristics (e.g., sex, birth date, birth weight in grams, and GA in weeks), and delivery hospital (e.g., locations and hospital type). After excluding stillbirths (death of the fetus before or during delivery after 22 weeks of gestation[32], $n = 30,307$), neonatal deaths (death within the first 28 days of life[33], $n = 6119$), and cases with undetermined sex (due to abnormal physiological development, $n = 1929$) and unmatched exposure ($n = 8747$), 4,021,741 live births were finally included in our analyses (Fig. 1).

### Outcome definitions
The primary outcome variable in this study was LBW, defined as birth weight below 2500 grams, regardless of GA. GA was ascertained by the number of weeks from the mother's last menstrual period to the birth date of the infant. To eliminate the occurrence of LBW due to fetal undergrowth in short gestation, we also included TLBW (birth weight below 2500 grams for pregnancies with at least 37 completed weeks of gestation) as a secondary endpoint outcome.

### Greenness exposure assessment
We utilized NDVI and EVI, two indicators representing the density of vegetation, to estimate maternal exposure to greenness. NDVI, the best-known and most commonly used vegetation indicator[16], allows easy and rapid identification of vegetation areas but is sensitive to aerosols and clouds. EVI is then developed as an optimized vegetation index, correcting for canopy background noise, aerosol conditions and soil effects. Thus, we used both NDVI and EVI as exposure indicators to perform the association analysis.

We extracted these two vegetation indices from imagery generated by NASA's Terra Satellite, the Moderate Resolution Imaging Spectroradiometer (MODIS) with a spatiotemporal resolution of 16 days and 250 × 250 m. Both indices range from −1 to 1, with values close to zero generally representing bare areas, and higher positive values indicating a higher density of vegetation[34–36]. We applied the maximum value composite method to rebuild the maximum vegetation index for each pregnant woman throughout her pregnancy. This method could well eliminate the influence of cloud cover and seasons, and better reflect the spatial distribution of green space[37]. Negative

values representing water bodies were not removed and were treated as zero prior to the assessment of greenness exposures[38].

To capture both local and larger-scale green spaces, NDVI and EVI within 500-m, 1000-m, 2000-m and 3000-m buffers around the maternal delivery hospitals were chosen to estimate greenness exposures during the entire pregnancy. These buffer sizes represented 10-, 15-, 20-, and 30-min walking distances from the delivery hospital, respectively[39]. Analyses based on multiple buffers were performed to validate the robustness of our findings.

### Covariates
In light of previous studies on greenness and birth outcomes[4], we considered several sets of covariates in our analysis: (1) maternal demographic characteristics: age, education attainment (below high school, high school, college or above), residence (city or village), nationality (Iranian or non-Iranian), delivery type (cesarean section or vaginal delivery), parity (0 for primipara, ≥1 for multipara), number of fetuses (singleton or multiple gestations) and delivery hospital; (2) fetal variables: infant sex (male or female), the season of birth (spring, summer, autumn, or winter), and GA; (3) environmental factors: average temperature, relative humidity, and $PM_{2.5}$ concentrations.

Average temperatures and dew point temperatures during the pregnancy period were aggregated from daily gridded estimates (0.1° × 0.1°) derived from a reanalysis dataset of the global climate (fifth generation, ERA5), which has been developed by the European Centre for Medium-Range Weather Forecast[40]. Relative humidity was then calculated using the method recommended by the National Weather Service[41]. We extracted monthly ground-level $PM_{2.5}$ concentration at a 0.1° × 0.1° resolution from the global surface $PM_{2.5}$ datasets released by the Atmospheric Composition Analysis Group[42,43]. Exposure levels of ambient temperature, relative humidity, and $PM_{2.5}$ for each pregnant woman throughout her pregnancy were consistently evaluated by mean values during months from the last menstrual period to the date of birth. Due to the minimal proportions of missing values for some socioeconomic variables, we elected to impute these sparse missing values to the mode of corresponding categorical variables in order to ensure modeling convergence[28].

### Association analysis
We conducted logistic regression models to quantify the association of greenness exposures with LBW/TLBW, adjusting for the confounders mentioned above. A sequential adjustment approach was adopted to determine the models with different adjustment levels. To be specific, Model 1 was a crude model without covariate adjustment; Model 2 was adjusted for maternal age, infant sex and GA; Model 3 was adjusted for covariates in maternal demographic characteristics and other fetal variables; Model 4 (fully adjusted model) was adjusted for covariates in maternal demographic characteristics, fetal variables, and environmental factors. Subsequent analyses were all performed based on a fully adjusted model. Considering the possible nonlinear relationships between birth outcomes and climate conditions, we incorporated temperature and relative humidity in our fully adjusted model using natural spline functions with three degrees of freedom[12,44].

We assessed ORs of LBW/TLBW with corresponding 95% CIs for each 0.1-unit increase in NDVI or EVI and for the second through fourth quartile exposure (Q2, Q3, and Q4) using the first quartile (Q1) as a reference. Tests for linear trends across quartiles were performed by introducing the median value of each quartile of NDVI or EVI as a continuous variable in the logistic regression model[45]. Since the linear assumptions between greenness exposure and birth outcomes may not hold, we replaced the linear term of NDVI or EVI with a restricted cubic spline (RCS) term with three knots placed at the 10th, 50th, and 90th percentiles of greenness exposure distribution to examine potential nonlinear relationships according to prior studies[10,46]. Likelihood-ratio test was used to compare the goodness of fit between

linear and RCS models, where $P$ value < 0.05 indicates a significant violation of the assumption of a linear relationship[47]. Subgroup analyses were done to evaluate potential effect modifiers, stratified by maternal age (≤25, 26–35, >35 years), education attainment (below high school, high school, college or above), and residence (city, village). Effect heterogeneity between subgroups was examined through a fixed effect meta-regression method[48].

Several sensitivity analyses were performed to test the robustness of our findings. First, the models were rerun after excluding records with birth weights <500 and >5000 grams to eliminate the confounding effect of extreme observations[49]. Second, records of live births involving multiple gestations were removed considering the influence of insufficient intrauterine growth induced by multiple births. Third, we excluded pregnant women under 13 years old and over 50 years old to reduce the likelihood of confounding by abnormal age of conception[50,51]. Fourth, analyses were repeated by excluding pregnant women with pre-existing chronic diseases (e.g., diabetes and chronic hypertension) or maternal delivery complications. Fifth, we additionally adjusted several provincial-level SES variables (i.e., gross domestic product per capita and rates of unemployment and medical insurance participation) in our fully-adjusted model, to partially account for potential unmeasured confounding. These SES variables were collected from the Iran Statistical Yearbook.

### Estimation of avoidable LBW under causal framework

Following the framework of counterfactual analysis widely adopted in prior modeling studies[26,27], avoidable LBW numbers due to greenness improvement were estimated by linking the E–R relationships to spatially resolved estimates of live births and maternal greenness exposures. Briefly, we predicted the risk of LBW/TLBW associated with greenness exposure with an interval of 0.01 unit, using a counterfactual exposure of mean value in given vegetation indices (NDVI/EVI) and buffer sizes (e.g., 500-m, 1000-m, 2000-m, and 3000-m) as the reference.

Considering the spatial variations in greenness and birth population, we initially resampled the annual maximum vegetation data of a 0.25-km resolution into a 1-km resolution using the nearest neighbor sampling method, to match the gridded estimates of live births in Iran for the year 2015 generated by the WorldPop[52]. We then calculated the ANs of LBW/TLBW births for each $1 \times 1$-km spatial unit using the Eqs. (1) and (2):

$$RR_s = \frac{OR_s}{(1-P)+P \times OR_s} \tag{1}$$

$$AN_s = W_s \times P \times AF_s = W_s \times P \times \frac{RR_s - 1}{RR_s} \tag{2}$$

where s refers to the spatial cell. $OR_s$ represents the point OR estimate derived from E–R functions for LBW/TLBW and a given exposure (defined by vegetation index and buffer) in $s^{th}$ spatial cell, referring to the greenness target ($GT_{mean}$, defined as mean NDVI/EVI for all spatial cells). $OR_s$ for spatial cells with greenspace > $GT_{mean}$ are defined as 1, suggesting no excess risks associated with poor greenspace. To avoid an overestimation[53], $OR_s$ is transformed into relative risk in $s^{th}$ spatial cell, denoted as $RR_s$. $P$ indicates the overall prevalence of LBW/TLBW estimated using birth records of the present study in 2015. $AN_s$ and $AF_s$ refer to avoidable numbers of LBW/TLBW and the attributable fractions in $s^{th}$ spatial cell, respectively. $W_s$ is gridded estimates of live births in $s^{th}$ spatial cell.

By summing up the ANs from each spatial cell, we yielded the national estimate of AN ($AN_{overall}$) in Iran for the year 2015. The overall AF at the national level ($AF_{overall}$) could be calculated through the equation $AF_{overall} = AN_{overall}/(\sum_s W_s \times P)$. The 95% uncertainty intervals for $AN_{overall}$ and $AF_{overall}$ were similarly generated using the

aforementioned equations, via substituting the point risk estimates with its lower and upper bounds derived from the E-R functions for given exposures.

All the analyses were performed in R software version 4.0.2 (R Foundation for Statistical Computing, Vienna, Austria). The *rms* (version 6.2-0), *mvmeta* (version 1.0.3), and *raster* (version 3.5–15) packages were applied to perform analyses for E-R association, meta-regression approach, and grid resampling, respectively. Figures were generated using the *ggplot2* (version 3.3.6) package. A two-sided *P*-value of <0.05 was considered statistically significant.

### Reporting summary

Further information on research design is available in the Nature Portfolio Reporting Summary linked to this article.

## Data availability

Satellite-derived datasets for NDVI and EVI are accessed at https://modis.gsfc.nasa.gov/data/dataprod/mod13.php. Meteorological factors are available at https://cds.climate.copernicus.eu/cdsapp#!/dataset/10.24381/cds.6c68c9bb?tab=overview. Ground-level estimates of PM$_{2.5}$ are obtained from https://sites.wustl.edu/acag/datasets/surface-pm2-5/. Gridded birth population data for Iran are derived from https://hub.worldpop.org/geodata/listing?id=18. Provincial-level SES variables are collected from https://www.amar.org.ir/english/Iran-Statistical-Yearbook. The birth cohort data are available under restricted access to protect the privacy of the study participants, access can be obtained by emailing corresponding authors YZ and FM, and requests will be responded to within 10 business days. Source data are provided with this paper.

## Code availability

The analytic and drawing codes are provided in the Supplementary Code. The relevant data to reproduce the figures are within Source Data.

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

## Acknowledgements

Dr. Yunquan Zhang was supported by the Youth Fund Project of Humanities and Social Sciences Research of the Ministry of Education (Grant No. 21YJCZH229), Hubei Provincial Natural Science Foundation of China (Grant No. 2021CFB032), and "The 14th Five Year Plan" Hubei Provincial Advanced Characteristic Disciplines (Groups) Project of Wuhan University of Science and Technology (Grant No. 2023C0102).

The authors would also like to acknowledge the support of CESAM (UIDP/50017/2020 + UIDB/50017/2020 + LA/P/0094/2020) and C2TN (UIDB/04349/2020). Thanks are due to FCT/MCTES for the contract granted to Helder Relvas (2021.00185.CEECIND).

## Author contributions

Y.Z. designed the research. F.M., H.R., M.B., and Y.Z. collected data. F.M. and Y.Z. cleaned data. S.L., Y.W., and Y.Z. contributed to performing statistical analyses, drafting the manuscript and interpreting the results. S.L., Y.W., K.W., Y.Y., Z.Y., and Y.Z. revised the manuscript. All authors read and approved the manuscript.

## Ethical approval

The birth cohort data were collected from the Neonatal Health Office of the Iranian Ministry of Iranian Health and Medical Education. The ethical issues of this study were approved by the Ethics Committee of Sabzevar University of Medical Sciences (IR.MEDSAB.REC.1396.99). Each pregnant woman signed written informed consent when she was admitted to the hospital.

## Competing interests

The authors declare no competing interests.
