## [Peer Review File · Nature Communications]

Surrounding greenness is associated with lower risk and burden of low birth weight in IranREVIEWER COMMENTS

Reviewer #1 (Remarks to the Author):

First, the paper could use some editing as evidenced by the following: “However, national evidence is still of wide lack in low- and middle-income countries experiencing high LBW prevalence.”

The paper could also use better definitions: What is the difference between stillbirth and fetal death? What do they mean by “vague fetal sex?” and why does it matter since they don’t use sex of baby to define low birthweight or growth restriction?

One of my concerns is that although this is clearly an observational study and can only provide evidence of an association, they imply cause and effect as shown in the title below. The authors need to go through the paper and restrict their conclusions to observations, not cause and effect.

“Greenness lowers risk and burden of low birth weight in Iran:

a national study of 4 million mother-infant pairs”

Another reason to be very cautious about their conclusions is recognized by the authors: “we did not consider parental characteristics before pregnancy (e.g., body-mass index, smoking, and drinking status) in our main analysis owing to data unavailability.” It is very likely that women living in less green areas have more of these characteristics.

I am troubled by the very large number of cases the authors calculated were avoidable with increasing greenness. Again since they only found an association, they need to be very cautious about these sort of claims.

I don’t know the birth distributions in Iran, but are 99% of the births first births? This does not seem to align with the Parity. This could be made clearer.

The above comments are of importance and need to be addressed, but I think the overall importance of a country-wide evaluation of greenness vs pregnancy outcome is potentially important. I wonder why, since there must be other outcomes available to study in this data base such as stillbirths, neonatal deaths, preterm births, etc., why the authors did not cast a wider net?

Reviewer #2 (Remarks to the Author):

In this study, the authors utilized a national dataset encompassing 4 million live birth records across 31 provinces in Iran, the authors investigated the risk and burden of low birth weight (LBW) associated with greenness exposure during pregnancy among maternity in Iran. This paper provided robust evidence on the beneficial effect of greenness in alleviating the risk and burden of low birth weight, suggesting that a greener environment may be a highly effective intervention strategy to address inequalities in LBW risk in low- and middle-income countries. The research topic was of broad interest, and the paper was generally well-written and well organized. Before accepting it for publication, I have several minor suggestions for authors' consideration.

Methods

1. Statistical methods for identifying differences between effect estimates of subgroups were not clarified, some more description of the method may be needed.
2. Could the authors provide the formula explicitly for the calculation of avoidable LBW numbers due to greenness improvement? I think it should be more clear for the readers to understand. The authors should provide some additional description on how they calculated the uncertainty (e.g., confidence interval) of the evaluation as shown in Figure 5 (error bar).
3. The R packages that the author used in main analyses should be listed in the method section.

Figures & Tables

4. Figure 3: The kernel density in the exposure-response relationship plot does not display the scale values.
5. The author excluded pregnant women with chronic diseases in the sensitivity analysis. Please provide the information on the prevalence of chronic diseases in Table 1.
6. The sum of some proportions did not equal 100% in Table 1, maybe due to the use of rounding-off method. Please check or provided some notes for clarity.
7. The word "PM2.5" in the title of the 27th reference is not subscript. Please kindly check the format of all references.

Reviewer #3 (Remarks to the Author):

A study with a novel idea has been conducted with a high sample size and strong analyses. I just have a few comments;

-If possible, reference should be given to the website from which the information of the two vegetation indices was extracted (line 119-121).

-Considering that the mother and birth data are from Iran which is not free access, please clarify in which university or scientific institution in Iran the project proposal has been approved? And which ethics committee issued the code of ethics. In addition, there is no explanation about the permission to access and using the registered data.

-Please add the ethical considerations section to the article.

-Please specify the type of study design in the method section.

RESPONSE TO REVIEWERS

Reviewers' Comments

Reviewer # 1 (Remarks to the Author)

1. First, the paper could use some editing as evidenced by the following: "However, national evidence is still of wide lack in low- and middle-income countries experiencing high LBW prevalence."

Response:

Thanks for your valuable suggestions. As suggested, we have thoroughly checked the text and used some necessary edits to improve its clarity and coherence. The snippet that reviewer mentioned was from the abstract section of the original manuscript. In our revised submission, we have simplified the Abstract section into a single paragraph using a concise way, so as to meet the requirements of the journal. The revised abstract could be seen as below and in our revised submission.

Revised Abstract

"Nexus between prenatal greenspace exposure and low birth weight (LBW) remains largely unstudied in low- and middle-income countries (LMICs). We investigated a nationwide retrospective cohort of 4021741 live births (263728 LBW births) across 31 provinces in Iran during 2013–2018. Greenness exposure during pregnancy was assessed using satellite-based normalized difference vegetation index (NDVI) and enhanced vegetation index (EVI). We estimated greenness-LBW associations using multiple logistic models, and quantified avoidable LBW cases under scenarios of improved greenspace through counterfactual analyses. Association analyses provided consistent evidence for approximately "L"-shaped exposure-response functions, linking 7.0–11.5% declines in the odds of LBW to each 0.1-unit rise in NDVI/EVI with multiple buffers. Assuming causality, 3931–5099 LBW births could be avoided by achieving greenness targets of mean NDVI/EVI, amounting to 4.4–5.6% of total LBW births in 2015. Our findings suggested potential health benefits of improved greenspace in lowering LBW risk and burden in LMICs." (Lines 24–36).

2. The paper could also use better definitions: What is the difference between stillbirth and fetal death? What do they mean by "vague fetal sex?" and why does it matter since they don't use sex of baby to define low birthweight or growth restriction?

Response:

Thanks for your suggestions. In this study, stillbirth refers to the death of the fetus before or during delivery after 22 weeks of gestation¹; fetal death actually refers to neonatal death, defined as death within the first 28 days of life². We apologize for our inappropriate choice of words and any confusion it may have caused. In the revised manuscript, we used "neonatal death" instead of "fetal death", and the definitions of stillbirth and neonatal death have been

added in the method section for clarity.

In this study, “vague fetal sex” refers to cases of the inability to determine the biological sex due to abnormal physiological development. The proportion of such cases is extremely low (1929/4068843), considered as an exceptional circumstance, they thus should not be included in the analysis. To avoid ambiguity, we have modified the wording and supplemented the explanation.

“After excluding stillbirths (death of the fetus before or during delivery after 22 weeks of gestation¹, n = 30307), neonatal deaths (death within the first 28 days of life², n = 6119), and cases with undetermined sex (due to abnormal physiological development, n = 1929) and unmatched exposure (n = 8747), 4021741 live births were finally included in our analyses.” (Lines 262–266).

3. One if my concern is that although this is clearly an observational study and can only provide evidence of an association, they imply cause and effect as shown in the title below. The authors need to go through the paper and restrict their conclusions to observations, not cause and effect. “Greenness lowers risk and burden of low birth weight in Iran: a national study of 4 million mother-infant pairs”.

Response:

Thanks for your insightful suggestions. We highly agree with the comment that our study is an observational study that can only provide an association between greenness and low birth weight. As suggested by the reviewer, we have modified the title to “Surrounding greenness is associated with lower risk and burden of low birth weight in Iran”. The entire text has been examined throughout, and we made some necessary edits to ensure that findings were limited to an association in both methods, results, and conclusions, by claiming “assuming causality and under counterfactual exposure framework”, or calling for “assessments from multiple countries utilizing causal analysis to validate findings from this study”.

For additional clarity, we have acknowledged the estimation of avoidable LBW on the basis of E-R functions derived from association analysis instead of causal framework as the fourth limitation of this study. Please check details for some revisions as below and in our revised submission.

“Under predefined causality and counterfactual scenarios of improved NDVI or EVI, we estimated greenness-related attributable fractions (AFs) and avoidable numbers (ANs) of LBW/TLBW by linking the E-R functions to spatially resolved estimates of live births, and maternal greenness exposures.” (Lines 121–124).

“Forth, our attributable assessments were on the basis of E-R functions derived from association analysis instead of causal framework, estimates of avoidable LBW should thus be interpreted with caution.” (Lines 238–240).

“Future assessments from multiple countries, utilizing sophisticated designs and causal analysis, are greatly warranted to validate findings from this study.” (Lines 250–252).

4. Another reason to be very cautious about their conclusions is recognized by the authors: “we did not consider parental characteristics before pregnancy (e.g., body-mass index, smoking, and drinking status) in our main analysis owing to data unavailability.” It is very likely that women living in less green areas have more of these characteristics.

Response:

Thanks for raising this concern. We greatly agree the reviewer’s comment that the interpretation of our finding should be very cautious. In the revised manuscript, we have emphasized in both the Results and Methods sections that our quantitative analysis was conducted on the basis of the framework of predefined causality and counterfactual exposure. In the interpretation of our results, we have made necessary adjustments to ensure that our findings do not extend beyond the realm of associations.

We also acknowledge that the potential influence of residual confounding such as maternal body-mass index and smoking status could, to some extent, affect the observed association between greenness exposure and LBW. Given that socioeconomic status (SES) might be associated with maternal body mass index and lifestyle, as a supplementary approach, we collected data on provincial-level SES variables including gross domestic product per capita, rates of unemployment and medical insurance participation from the Iran Statistical Yearbook. We conducted a sensitive analysis via additionally adjusting for these provincial SES variables as individual proxies to explore the impact of SES adjustment on greenness-LBW associations.

This sensitivity analysis yielded risk effects that were highly comparable to our primary findings (see the table below). Also, prior cohort evidence from Asia³ and Europe⁴ demonstrated comparable associations between greenness and LBW, when comparing unadjusted models to multivariable models adjusted for various covariates including lifestyle factors such as maternal smoking status. These analyses to some extent suggested that the potential impact of residual confounding due to lifestyle factors should be very limited on our findings. To clarify this point to the readers, we have updated the limitation of residual confounding in our discussion (lines 231–238), and have included the comparative analyses adjusted for several provincial-level SES variables (i.e., gross domestic product per capita and rates of unemployment and medical insurance participation) in our sensitivity analyses (Lines 352–355). Please check details for these revisions as below and in our revised submission.

“Third, unmeasured confounding (e.g., body-mass index and smoking status) was likely from the lack of control for potential individual-level risk factors of LBW. However, sensitive analyses via additionally adjusting for provincial-level SES variables provided comparable risk estimates with our primary results. Also, prior cohort studies from Asia³ and Europe⁴ reported similar greenness-LBW associations between unadjusted and multivariable-adjusted analyses adjusted for various covariates including lifestyle factors such as maternal smoking status. Such evidence suggested a limited influence of residual confounding on our findings.” (Lines 231–238).

“Fifth, we additionally adjusted several provincial-level SES variables (i.e., gross domestic product per capita and rates of unemployment and medical insurance participation) in our

multivariable-adjusted models, to partially account for potential unmeasured confounding. These SES variables were collected from Iran Statistical Yearbook.” (Lines 352–355).

Supplementary Table for sensitivity analyses

Buffers	LBW (OR, 95% CI)		TLBW (OR, 95% CI)	
	NDVI	EVI	NDVI	EVI
Fully-adjusted analyses in main text				
500 m	0.930 (0.927–0.933)	0.913 (0.910–0.917)	0.921 (0.918–0.924)	0.902 (0.898–0.907)
1000 m	0.923 (0.920–0.926)	0.904 (0.900–0.908)	0.913 (0.910–0.917)	0.892 (0.887–0.896)
2000 m	0.916 (0.914–0.919)	0.893 (0.890–0.897)	0.906 (0.902–0.909)	0.880 (0.876–0.885)
3000 m	0.911 (0.908–0.914)	0.885 (0.881–0.889)	0.899 (0.895–0.902)	0.871 (0.866–0.875)
Sensitive analyses additionally adjusted for provincial-level SES variables				
500 m	0.938 (0.935–0.941)	0.926 (0.922–0.930)	0.928 (0.925–0.932)	0.914 (0.910–0.919)
1000 m	0.932 (0.929–0.935)	0.918 (0.914–0.921)	0.921 (0.917–0.925)	0.904 (0.900–0.909)
2000 m	0.925 (0.922–0.928)	0.907 (0.903–0.911)	0.913 (0.909–0.917)	0.893 (0.888–0.898)
3000 m	0.919 (0.916–0.922)	0.899 (0.895–0.904)	0.906 (0.903–0.910)	0.884 (0.879–0.889)

Abbreviations: LBW, low birth weight; TLBW, term low birth weight; OR, odds ratio; CI, confidence interval; NDVI, normalized difference vegetation index; EVI, enhanced vegetation index; SES, socioeconomic status.

5. I am troubled by the very large number of cases the authors calculated were avoidable with increasing greenness. Again since they only found an association, they need to be very cautious about these sort of claims.

Response:

Thanks for your thoughtful feedback and apologized for this confusing estimation. While checking the attributable analyses in our original manuscript, the avoidable LBW cases were simply calculated at the provincial scale, which did not take into account the spatial variations in greenness and birth population at finer scales. This crude estimation should have largely overestimated the greenness-related burden, owing to the ignorance of spatial units that have already achieved the greenness target. Great thanks for bringing this concern to our caution and for providing us an opportunity to improve the analysis.

In our revised submission, we have refined the attributable analyses at a spatially resolved scale, so as to fully capture the spatial variations of exposure and population. Specifically, in line with gridded estimates of live births at a 1-km resolution generated by the WorldPop⁵, we calculated avoidable LBW cases for each 1 × 1-km grid scale through linking exposure-response functions with LBW births and greenness exposure. By summing up the cases of each grid, we could obtain the national estimate of total avoidable LBW number. Such an approach has been widely adopted in health burden assessments attributable to

environmental risk exposure (e.g., green space and fine particular matter)⁶⁻⁸. Details for equations and calculation methods were described in the updated Method section “Estimation of avoidable LBW under causal framework”. The avoidable LBW number (3931–5099) and attributable fraction (4.4–5.6%) within multiple buffers have been updated in our resubmission (Figure 5), and we again apologize for oversight in our original manuscript. As the reviewer suggested, in our revised submission, we also made some edits when interpreting these analyses with caution by claiming “assuming causality and under counterfactual exposure framework”. Please check details for some revisions as below and in our revised submission.

Figure 5 Estimates of attributable fraction (A) and avoidable numbers (B) of LBW and TLBW across Iran in 2015 by achieving greenness targets of mean NDVI/EVI within multiple buffers.

6. I don't know the birth distributions in Iran, but are 99% of the births first births? This does not seem to align with the Parity. This could be made clearer.

Response:

Thanks for your helpful suggestions. The inappropriate use of “birth order” in our descriptive table may have mislead the reviewers. As indicated in Table 1, the first-born births accounted for 41.5% (Parity = 0) of all live births in this study. Parity is a binary variable (0 vs. ≥1), where 0 indicates primipara (first-time mother), and ≥1 indicates multipara (woman who has given birth before).

The estimate of 99% mentioned by the reviewer actually represented the proportion of singleton fetuses, as indicated by binary variable of “Birth order” in Table 1, where our original intention is to refer “0” for singleton and “≥1” for multifetation. We apologize for this inappropriate term and any confusion it may have caused. In the revised manuscript, we used “fetal number (singleton vs. multifetation)” instead of “birth order (0 vs. ≥1)”. For clarity, we have included some notes in the Covariates section.

“parity (0 for primipara, ≥1 for multipara), fetal number (singleton or multifetation).” (Line 301).

7. The above comments are of importance and need to be addressed, but I think the overall importance of a country-wide evaluation of greenness vs pregnancy outcome is potentially important. I wonder why, since there must be other outcomes available to study in this data base such as stillbirths, neonatal deaths, preterm births, etc., why the authors did not cast a wider net?

Response:

Thanks for your insightful feedback. The comments raised above by the reviewer are indeed important and should be addressed in our study. We appreciate your great interests in the broader scope of this research. While we highly agree that country-wide assessments of greenness versus other pregnancy outcomes (e.g., stillbirths, neonatal deaths, and preterm births) should be also of great value, this study provided a focused analysis on LBW only rather than all together, primarily for the purpose to avoid the reduced interpretability of our findings on greenspace-associated risk and burden, given that no strong evidence was available worldwide to date supporting the nexus with neonatal deaths and stillbirths (lack unified definitions worldwide). In addition, the focus on LBW in less developed locations echoes with its great implications on maternal and fetal health from a life course perspective, as well as the World Health Organization Global Nutrition Targets 2025 for LBW—a 30% reduction in the number of LBW babies by 2025, as noted in the introduction.

Again, we appreciate reviewer’s suggestion and will consider incorporating additional outcomes in future assessments. Expanding the scope could indeed provide a more comprehensive understanding of the relationship between greenness and pregnancy outcomes, particularly in low- and middle-income countries. Thanks for bringing this to our attention, and we will take your input into account for future research endeavors.

Reviewer # 2 (Remarks to the Author)

General comments

In this study, the authors utilized a national dataset encompassing 4 million live birth records across 31 provinces in Iran, the authors investigated the risk and burden of low birth weight (LBW) associated with greenness exposure during pregnancy among maternity in Iran. This paper provided robust evidence on the beneficial effect of greenness in alleviating the risk and burden of low birth weight, suggesting that a greener environment may be a highly effective intervention strategy to address inequalities in LBW risk in low- and middle-income

countries. The research topic was of broad interest, and the paper was generally well-written and well organized. Before accepting it for publication, I have several minor suggestions for authors' consideration.

Response:

Thanks for your careful work and comments on our submission. We have carefully revised our manuscript based on your point-by-point comments, and provided some explanations for your questions. Hope our revisions and explanations could be in your favor.

1. Methods: Statistical methods for identifying differences between effect estimates of subgroups were not clarified, some more description of the method may be needed.

Response:

Thanks for your kind reminder. Effect heterogeneity between subgroups was examined through a meta-regression method well adopted in environmental health studies, and we have added this description in revised manuscript.

“Effect heterogeneity between subgroups was examined through a fixed effect meta-regression method⁹” (Lines 341–343)

2. Could the authors provide the formula explicitly for the calculation of avoidable LBW numbers due to greenness improvement? I think it should be more clear for the readers to understand. The authors should provide some additional description on how they calculated the uncertainty (e.g., confidence interval) of the evaluation as shown in Figure 5 (error bar).

Response:

Thanks for your thoughtful feedback on our manuscript. We have provided the computational formulas and included a detailed description of the formula used. Additionally, we have included a necessary explanation of the uncertainty estimation for attributable analyses in Figure 5 in the revised manuscript. Motivated by the reviewer's suggestions, we have restructured the paragraphs to enhance coherence and clarity.

“Considering the spatial variations in greenness and birth population, we initially resampled the annual maximum vegetation data of a 0.25-km resolution into a 1-km resolution using the nearest neighbor sampling method, to match the gridded estimates of live births in Iran for the year 2015 generated by the WorldPop⁵. We then calculated the ANs of LBW/TLBW births for each 1 × 1-km spatial unit using the equations (1) & (2):

$$RR_s = \frac{OR_s}{(1 - P) + P \times OR_s} \quad (1)$$

$$AN_s = W_s \times P \times AF_s = W_s \times P \times \frac{RR_s - 1}{RR_s} \quad (2)$$

Where s refers to the spatial cell. OR_s represents the point OR estimate derived from E-R functions for LBW/TLBW and a given exposure (defined by vegetation index and buffer) in s^{th} spatial cell, referring to the greenness target (GT_{mean} , defined as mean NDVI/EVI for all spatial cells). OR_s for spatial cells with greenspace $> GT_{\text{mean}}$ are defined as 1, suggesting no excess

risks associated with poor greenspace. To avoid an overestimation¹⁰, OR_s is transformed into relative risk in s^{th} spatial cell, denoted as RR_s . P indicates the overall prevalence of LBW/TLBW estimated using birth records of the present study in 2015. AN_s and AF_s refer to avoidable numbers of LBW/TLBW and the attributable fractions in s^{th} spatial cell, respectively. W_s is gridded estimates of live births in s^{th} spatial cell.

By summing up the ANs from each spatial cell, we yielded the national estimate of AN ($AN_{overall}$) in Iran for the year 2015. The overall AF at national level ($AF_{overall}$) could be calculated through the equation $AF_{overall} = AN_{overall} / (\sum_s W_s \times P)$. The 95% uncertainty intervals for $AN_{overall}$ and $AF_{overall}$ were similarly generated using the aforementioned equations, via substituting the point risk estimates with its lower and upper bounds derived from the E-R functions for given exposures.” (Lines 364–385).

3. The R packages that the author used in main analyses should be listed in the method section.

Response:

Thanks for your valuable suggestion. In the revised manuscript, we have provided a list of R packages used in our study and clarified its functions in our analyses. This will facilitate the reproducibility of our analysis and improve the clarity.

“The *rms* (version 6.2-0), *mvmeta* (version 1.0.3), and *raster* (version 3.5-15) packages were applied to perform analyses for E-R association, meta-regression approach, and grid resampling, respectively. Figures were generated using the *ggplot2* (version 3.3.6) package.” (Lines 387–390).

4. Figures & Tables: Figure 3: The kernel density in the exposure-response relationship plot does not display the scale values.

Response:

As the reviewer suggested, Figure 3 was updated to ensure that it now includes the scale values of both NDVI and EVI distributions. Also, we provided some details of text explanations at the bottom of the figure to enhance the clarity.

“The solid lines in the upper panels represent the point estimates of LBW/TLBW risks, while the shaded bands represent their 95% confidence intervals. In the lower panels, we have displayed the kernel density curves and boxplots to illustrate the distributions of NDVI and EVI exposures.” (Explanatory text for Figure 3).

5. The author excluded pregnant women with chronic diseases in the sensitivity analysis. Please provide the information on the prevalence of chronic diseases in Table 1.

Response:

Thanks for your advice. The prevalence of maternal diabetes and hypertension in this study was 2.5% and 1.5%, respectively, and we have included this information in Table 1 of the revised submission.

6. The sum of some proportions did not equal 100% in Table 1, maybe due to the use of

rounding-off method. Please check or provided some notes for clarity.

Response:

Thanks for your careful work. We have carefully checked Table 1 and found that the proportional sum of certain variables is not precisely 100% due to rounding-off methods, as the reviewer guessed. For clarity, we have provided a necessary note regarding this matter at the bottom of the table in our revised manuscript.

“The sum of percentages from multiple subgroups may not equal 100% exactly due to rounding-off numbers.” (Notes in Table 1).

7. The word "PM2.5" in the title of the 27th reference is not subscript. Please kindly check the format of all references.

Response:

We have thoroughly reviewed the format of all references and made necessary amendments in accordance with the guidelines of Nature Communications.

Reviewer # 3 (Remarks to the Author)

General comments

A study with a novel idea has been conducted with a high sample size and strong analyses. I just have a few comments.

Response:

Thanks for your time. We have accordingly revised the details of our manuscript based on your helpful suggestions. Hope our updates could be in your favor.

1. If possible, reference should be given to the website from which the information of the two vegetation indices was extracted (line 119-121).

Response:

Data and related information of the two vegetation indices can be accessed at <https://modis.gsfc.nasa.gov/>, and we have updated this information in the revised manuscript as suggested.

2. Considering that the mother and birth data are from Iran which is not free access, please clarify in which university or scientific institution in Iran the project proposal has been approved? And which ethics committee issued the code of ethics. In addition, there is no explanation about the permission to access and using the registered data. Please add the ethical considerations section to the article.

Response:

Thanks for your kind reminder. We have added the sections of Ethical approval and Data availability to the article, please check the updates as below and in our revised manuscript.

Ethical approval

This study was approved by the Ethics Committee of Sabzevar University of Medical Sciences (IR.MEDSAB.REC.1396.99)." (Lines 392–394).

Data availability

Satellite-derived datasets for NDVI and EVI are accessed at <https://modis.gsfc.nasa.gov/>. Meteorological factors are available at <https://cds.climate.copernicus.eu/>. Ground-level estimates of PM_{2.5} are obtained from <http://sites.wustl.edu/acag/>. Gridded birth population data for Iran are derived from <http://www.worldpop.org.uk/data/>. Provincial-level SES variables are collected from <https://www.amar.org.ir/english/>. The birth cohort data that support the findings of this study are available from the corresponding authors upon reasonable request." (Lines 395–402).

3. Please specify the type of study design in the method section.

Response:

As the reviewer suggested, we have clarified this study design (retrospective birth cohort) in the Method section.

"This nationwide retrospective birth cohort included 4068843 birth records from 749 hospitals (Figure S5) across 31 Iranian provinces between January 2013 and December 2018." (Lines 255–256).

References

- 1 Lawn, J. E. *et al.* Stillbirths: rates, risk factors, and acceleration towards 2030. *Lancet* **387**, 587-603, doi:10.1016/S0140-6736(15)00837-5 (2016).
- 2 Cnattingius, S., Johansson, S. & Razaz, N. Apgar Score and Risk of Neonatal Death among Preterm Infants. *N Engl J Med* **383**, 49-57, doi:10.1056/NEJMoa1915075 (2020).
- 3 Lee, P. C. *et al.* Residential greenness and birth outcomes: Evaluating the mediation and interaction effects of particulate air pollution. *Ecotoxicol Environ Saf* **211**, 111915, doi:10.1016/j.ecoenv.2021.111915 (2021).
- 4 Grazuleviciene, R. *et al.* Surrounding greenness, proximity to city parks and pregnancy outcomes in Kaunas cohort study. *Int J Hyg Environ Health* **218**, 358-365, doi:10.1016/j.ijheh.2015.02.004 (2015).
- 5 James, W. H. M. *et al.* Gridded birth and pregnancy datasets for Africa, Latin America and the Caribbean. *Sci Data* **5**, 180090, doi:10.1038/sdata.2018.90 (2018).
- 6 Xue, T. *et al.* Estimation of stillbirths attributable to ambient fine particles in 137 countries. *Nat Commun* **13**, 6950, doi:10.1038/s41467-022-34250-4 (2022).
- 7 Barboza, E. P. *et al.* Green space and mortality in European cities: a health impact assessment study. *Lancet Planet Health* **5**, e718-e730, doi:10.1016/S2542-5196(21)00229-1 (2021).
- 8 Southerland, V. A. *et al.* Global urban temporal trends in fine particulate matter (PM_{2.5}) and attributable health burdens: estimates from global datasets. *Lancet Planet Health* **6**, e139-e146, doi:10.1016/S2542-5196(21)00350-8 (2022).
- 9 Ye, T. *et al.* Short-term exposure to wildfire-related PM_{2.5} increases mortality risks and

- burdens in Brazil. *Nat Commun* **13**, 7651, doi:10.1038/s41467-022-35326-x (2022).
- 10 Zhang, J. & Yu, K. F. What's the relative risk? A method of correcting the odds ratio in cohort studies of common outcomes. *JAMA* **280**, 1690-1691, doi:10.1001/jama.280.19.1690 (1998).

REVIEWERS' COMMENTS

Reviewer #4 (Remarks to the Author):

Thank you for your reply. All the questions used have been well answered, and the manuscript is suitable for publication in this magazine.

Reviewer #5 (Remarks to the Author):

My previous comments have been modified and I have no new comments.

Reviewer #6 (Remarks to the Author):

Siqi Luo et al. Surrounding greenness is associated with lower risk and burden of low birth weight in Iran

This interesting article combines national birth data on more than four million births with satellite-based vegetation information. The used methods are sound, the used data sources are valid, and the analyses are carefully performed. The main result is that a decline of 7 to 11.5% can be seen in the odds of low birth weight to each 0.1-unit rise in vegetation index calculated by the researchers. The authors conclude that improved greenspace does bring potential health benefits and may lead to lowering low birth weight risk.

The article has been revised thoroughly after a previous review, and the current version clearly states that the found observation is an association, and the causality cannot be confirmed. The calculations of avoided number of low-birth weight children are thus theoretical, but can be kept in the article to show the effect size.

The strength of this paper is the well advanced statistical and data science methods as well as rich register data. The use of regional SES data is a good addition, even though the main analyses included individual-based data on parturients' socioeconomic background. The main weakness is that the study is an observational cross-sectional study. However, the weaknesses are discussed in the article.

I suggest that this paper can be accepted for publication after some language and other improvements.

- Add thousand separators in large numbers to ease the reading.
- The quotation marks are not necessary in words as L-shaped, U-shaped, and S-pattern.
- Fetal number could be replaced by number of fetuses.
- There are still some room for language improvement, for example the statement 'do a poor job' and the terms 'non-singleton' and 'multifetation' could be revised.
- Add the relevant definitions and/or ICD-codes in Tables, for example for delivery complication, gestational diabetes and gestational hypertensions.

RESPONSE TO REVIEWERS

Journal: *Nature Communication*

Manuscript ID: NCOMMS-23-09249A

Decision Received: Sep 25, 2023

Revision (A) Submitted: Oct 8, 2023

Dear editors and reviewers,

Thank you for giving us the opportunity to submit a revised draft of the manuscript "Surrounding greenness is associated with lower risk and burden of low birth weight in Iran". We really appreciate the time and effort that you dedicated to improving our paper and are grateful for the valuable comments and suggestions. We have read all comments and suggestions deliberately and made necessary revisions and explanations accordingly. The revision notes are as follows:

- (1) *Reviewers' comments are in blue italic type.*
- (2) Authors' responses are in black normal font.
- (3) Revisions on manuscript are highlighted by yellow color.

Reviewer's Comments

Reviewer #4 (Remarks to the Author):

Thank you for your reply. All the questions used have been well answered, and the manuscript is suitable for publication in this magazine.

Response: We thank the reviewer for your time on reviewing this paper.

Reviewer #5 (Remarks to the Author):

My previous comments have been modified and I have no new comments.

Response: We thank the reviewer for your time on reviewing this paper.

Reviewer #6 (Remarks to the Author):

Siqi Luo et al. Surrounding greenness is associated with lower risk and burden of low birth weight in Iran

This interesting article combines national birth data on more than four million births with satellite-based vegetation information. The used methods are sound, the used data sources are valid, and the analyses are carefully performed. The main result is that a decline of 7 to 11.5% can be seen in the odds of low birth weight to each 0.1-unit rise in vegetation index calculated by the researchers. The authors conclude that improved greenspace does bring potential health benefits and may lead to lowering low birth weight risk.

The article has been revised thoroughly after a previous review, and the current version clearly states that the found observation is an association, and the causality cannot be confirmed. The calculations of avoided number of low-birth weight children are thus theoretical, but can be kept in the article to show the effect size.

The strength of this paper is the well advanced statistical and data science methods as well as rich register data. The use of regional SES data is a good addition, even though the main analyses included individual-based data on parturients' socioeconomic background. The main weakness is that the study is an observational cross-sectional study. However, the weaknesses are discussed in the article.

I suggest that this paper can be accepted for publication after some language and other improvements.

Response:

Thanks for your careful work and comments on our submission. We have carefully revised our manuscript based on your point-by-point comments. Hope our revisions and explanations could be in your favor.

1. Add thousand separators in large numbers to ease the reading.

Response: Thousand separators have been added to large numbers to enhance readability, please check our revised submission for details.

2. The quotation marks are not necessary in words as L-shaped, U-shaped, and S-pattern.

Response: Quotation marks have been omitted from words such as L-shaped, U-shaped, and S-pattern.

3. Fetal number could be replaced by number of fetuses.

Response: The term "Fetal number" has been replaced with "Number of fetuses" for clarity and precision.

4. There are still some room for language improvement, for example the statement 'do a poor job' and the terms 'non-singleton' and 'multifetation' could be revised.

Response: Done. The phrase "do a poor job" has been revised for more appropriate language. Terminology adjustments have been made: "non-singleton" and "multifetation" has been refined to "multiple gestation".

"Second, NDVI and EVI may exhibit limitations in fully characterizing the maternal experience of being exposed to trees and plants, because these indicators do not reflect the type and quality of vegetation." (Line 227–229)

"Second, records of live births involving multiple gestations were removed considering the influence of insufficient intrauterine growth induced by multiple births." (Line 346–348)

5. Add the relevant definitions and/or ICD-codes in Tables, for example for delivery complication, gestational diabetes and gestational hypertensions.

Response: Sorry for our miss. The variables about chronic disease refers to "chronic hypertension" and "diabetes" rather than "gestational hypertension" and "gestational diabetes" as originally mentioned in the table 1. We apologize for any confusion caused by this oversight. As you suggested, we have labeled the relevant definitions for "chronic hypertension", "diabetes", and "delivery complication" below Table 1.

"Delivery complications referred to a range of medical issues and problems that can occur during the process of delivery, including one or more injuries of the parturient canal, placenta abruption, premature rupture of membranes, shoulder dystocia, and uterine rupture at delivery, etc. Diabetes was defined as type 1 or type 2 diabetes diagnosed prior to conception (without gestational diabetes). Chronic hypertension was defined as pre-pregnancy hypertension ($\geq 140/90$ mmHg) or hypertension before 20 weeks of pregnancy." (below Table 1)